# The Mitochondrial Antioxidant Sirtuin3 Cooperates with Lipid Metabolism to Safeguard Neurogenesis in Aging and Depression

**DOI:** 10.3390/cells11010090

**Published:** 2021-12-29

**Authors:** Sónia Sá Santos, João B. Moreira, Márcia Costa, Rui S. Rodrigues, Ana M. Sebastião, Sara Xapelli, Susana Solá

**Affiliations:** 1Research Institute for Medicines (iMed.ULisboa), Faculty of Pharmacy, Universidade de Lisboa, 1649-003 Lisbon, Portugal; joaomoreira@medicina.ulisboa.pt (J.B.M.); mafi.mafi@gmail.com (M.C.); 2Instituto de Medicina Molecular (iMM) João Lobo Antunes, Faculdade de Medicina, Universidade de Lisboa, 1649-028 Lisbon, Portugal; rmsrodrigues@medicina.ulisboa.pt (R.S.R.); anaseb@medicina.ulisboa.pt (A.M.S.); sxapelli@medicina.ulisboa.pt (S.X.); 3Department of Translational Neurodegeneration, German Center for Neurodegenerative Diseases (DZNE), 81377 Munich, Germany; 4Instituto de Farmacologia e Neurociências, Faculdade de Medicina, Universidade de Lisboa, 1649-028 Lisbon, Portugal

**Keywords:** aging, depression, lipid metabolism, mitochondria, neural stem cells, SIRT3

## Abstract

Neural stem cells (NSCs), crucial for memory in the adult brain, are also pivotal to buffer depressive behavior. However, the mechanisms underlying the boost in NSC activity throughout life are still largely undiscovered. Here, we aimed to explore the role of deacetylase Sirtuin 3 (SIRT3), a central player in mitochondrial metabolism and oxidative protection, in the fate of NSC under aging and depression-like contexts. We showed that chronic treatment with tert-butyl hydroperoxide induces NSC aging, markedly reducing SIRT3 protein. SIRT3 overexpression, in turn, restored mitochondrial oxidative stress and the differentiation potential of aged NSCs. Notably, SIRT3 was also shown to physically interact with the long chain acyl-CoA dehydrogenase (LCAD) in NSCs and to require its activation to prevent age-impaired neurogenesis. Finally, the SIRT3 regulatory network was investigated in vivo using the unpredictable chronic mild stress (uCMS) paradigm to mimic depressive-like behavior in mice. Interestingly, uCMS mice presented lower levels of neurogenesis and LCAD expression in the same neurogenic niches, being significantly rescued by physical exercise, a well-known upregulator of SIRT3 and lipid metabolism. Our results suggest that targeting NSC metabolism, namely through SIRT3, might be a suitable promising strategy to delay NSC aging and confer stress resilience.

## 1. Introduction

Adult neurogenesis constitutively occurs in the adult mammalian brain where neural stem cells (NSCs) are able to differentiate into three neural lineages, including neurons, astrocytes, and oligodendrocytes [1]. Moreover, these multipotent cells are also able to self-renew through cell proliferation in order to maintain their pool [2]. This process occurs mainly in two restricted brain areas, the subventricular zone (SVZ) lining the lateral ventricles, and the subgranular zone (SGZ) within the dentate gyrus (DG) of the hippocampus [3,4]. In fact, these regions are rich in NSCs that originate neuroblasts, which then migrate toward their final destinations, where they complete their differentiation into mature neurons and are integrated into the neuronal circuitry [5,6,7].

Neurogenesis is a highly complex process of generating functional neurons from NSCs and comprises several steps: proliferation and neuronal fate specification, migration, and the differentiation and survival of the newly formed neurons, with functional integration in the existing neuronal circuits [8]. Due to its immense intricacy, this process occurs under tight regulation by a plethora of regulatory mechanisms [4]. There are several factors that contribute to the systemic regulation of adult neurogenesis. One of the most important is age. In fact, the number of NSCs, as well as their potential for self-renewal and differentiation, sharply drops throughout the age of the organism [9,10]. NSCs are therefore particularly responsive to several stimuli present in the aging process, including oxidative stress, mitochondrial dysfunction, and a compromised bioenergetic metabolism [11,12]. Accordingly, the progressive decrease in NSCs’ ability to self-renew and differentiate into new neurons throughout aging ends up reflected in the progressive loss of brain physiological integrity and in the lower regenerative ability that follows the aging process [9,10,13].

In addition, recent reports have also shown that NSC decline is a risk factor for psychiatric disorders, including anxiety and depression [14,15]. Notably, depression has been shown to be the most common type of mental illness in adulthood. It is commonly encountered with dementia in the elderly and the correlation between both neurological problems has become increasingly evident [16,17]. The major depressive disorder has been associated with decreased levels of neural markers, expressed by both NSCs and mature neurons, as well as with a lower neurogenic niche volume [18]. In fact, previous evidence shows that anxiety- and depression-like states lead to impairments in adult subventricular neurogenesis [18]. Nevertheless, the vast majority of studies have been correlative, and there are several gaps in our understanding of the molecular mechanisms responsible for this strong association between physiological neurogenesis and individual wellbeing.

Although the underlying molecular mechanisms of adult neurogenesis are still not fully unraveled, mitochondria have been shown to be a key regulator of the fate of NSCs [19,20]. Indeed, the role of mitochondria activity in NSC biology has been dissected in numerous recent studies, which in turn have suggested that mitochondria bioenergetic and biogenesis are driving forces in NSC regulation and adult neurogenesis [19,21,22]. In particular, we and others have demonstrated that metabolic plasticity, including the subtle balance between lipogenesis and lipid catabolism, is crucial to the transition between stemness maintenance and commitment to differentiation [23,24]. Notably, a reduction in the mitochondrial content and a lower oxidative metabolism have been reported in aged NSCs [25] as well as a metabolic dysfunction in psychiatric-disorder-affected brains postmortem [26]. 

The mitochondrial deacetylase Sirtuin 3 (SIRT3) is a central player in mitochondrial metabolism and oxidative protection [27,28,29] often associated with aging [28]. Aside from histones, SIRTs target other proteins in the cytoplasm and mitochondria by post-translationally removing acetyl groups attached to their lysine residues by acetyltransferases [30]. SIRT3 exhibits a nuclear and mitochondrial localization and is responsible for major mitochondrial deacetylase activity, having a profound contribution in mitochondrial biology [31]. Evidence shows that SIRT3 is involved in mitochondrial dynamics and biogenesis, ATP generation, mitochondrial unfolded protein response (mtUPR), and cell death signaling [28,29]. More frequently, SIRT3 is known for influencing mitochondrial reactive oxygen species (ROS) homeostasis, namely by increasing the activation of superoxide dismutase 2 (SOD2)-deacetylase, a major superoxide-scavenger in mitochondria [32]. The acetylation status of SOD2 lysine 68 inversely correlates with its antioxidant activity, suggesting a post-translational regulatory mechanism by means of acetylation [33]. Thus, SIRT3 participates in SOD2 fine-tuning through the removal of the acetyl group therefore increasing its activity and cell survival [33]. Apart from the regulatory effect on the mitochondrial oxidative state [34], mitochondrial SIRT3 also influences energy metabolism and nutrient-sensing pathways. It has been shown that SIRT3 can interact with pyruvate dehydrogenase, regulating its acetylation/deacetylation status, and therefore having a preponderant regulatory role in glycolysis [35]. Additionally, SIRT3 is able to promote fatty acid (FA) catabolism and the use of triacylglycerols by deacetylating and activating long chain acyl-CoA dehydrogenase (LCAD) [36]. LCAD is a key enzyme in mitochondrial β-oxidation, a catabolic process responsible for the degradation of short, medium, and long chain FAs to acetyl-CoA [37]. SIRT3 promotes fatty acid β-oxidation by directly deacetylating LCAD at lysine 42 thus increasing its activity and promoting mitochondrial β-oxidation [36]. 

Although the impact of mitochondria in NSC fate has been well-established, the precise role of mitochondrial SIRT3 state in mediating NSC function is still not fully understood. Dissecting the mechanistic connection between key metabolic regulators and NSC biology might provide a broader perspective for understanding the processes of adult neurogenesis, and thus novel opportunities to design new and suitable strategies to boost NSC-driven regeneration in conditions associated with a decline in neurogenesis, such as aging and depression. Hence, here we sought to explore the molecular mechanisms underlying SIRT3’s regulation of the fate of NSCs, and whether SIRT3 and its downstream targets would be good pharmacological targets to rescue neurogenic potential in aging and depression contexts.

## 2. Materials and Methods

### 2.1. Neural Stem Cell Cultures

The CGR8-NS cells were derived from embryonic stem cell line CGR8, established from the inner cell mass of a 3.5 day male pre-implantation mouse embryo (ECACC 07032901) [38]. This robust cellular model can be stably expanded in vitro and maintain neuronal and glial differentiation potential even after long-term passaging [39,40,41]. These cells were obtained from Prof. Smith’s laboratory, University of Cambridge, and kindly provided by Dr. Margarida Diogo, Universidade de Lisboa.

NSCs were grown in a monolayer and routinely maintained in self-renewal conditions in Euromed-N medium (EuroClone^®^ S.p.A., Milan, Italy), at 37 °C, in a humidified atmosphere of 5% CO_2_. The self-renewal medium was supplemented with 1% penicillin-streptomycin, 1% N-2 supplement, 20 ng/mL epidermal growth factor (EGF), and 20 ng/mL basic fibroblast growth factor (βFGF). Medium supplements were from Gibco™ (Thermo Fisher Scientific Inc., Waltham, MA, USA). All experiments were performed in self-renewal medium, and NSCs were plated onto uncoated tissue culture plastic dishes at a density of around 1 × 10^5^ cells/cm^2^.

### 2.2. Chronic Cell Aging Model

Chronic aging was induced in NSCs by sequential short-term exposure to the oxidative agent tert-butyl hydroperoxide (tBHP, Luperox^®^ TBH 70x, Sigma-Aldrich Corp., St. Louis, MO, USA), and the protocol was adapted as described elsewhere [42]. Chronic treatment started 22 h after plating, where NSCs were incubated with 50 μM tBHP for 2 h/day, for 4 consecutive days. In parallel, an equal volume of the vehicle was added to control cultures. After each 2 h incubation, all NSC cultures were washed with PBS (Gibco™) and then rested in self-renewal medium for 22 h. At the end of the fourth tBHP-treatment (or vehicle for controls) NSC cells were detached with StemPro^®^ Accutase^®^ (Gibco™) and processed for further analysis.

### 2.3. Evaluation of Cell Death and Viability

Viability and cell death levels were assessed by Guava Nexin^®^ reagent (4500-0450; Luminex Corp., Austin, TX, USA), according to the manufacturer’s instructions. At the end of the chronic tBHP regimen, the culture medium containing death cells was collected together with adherent NSCs detached with StemPro^®^ Accutase^®^, centrifuged for 5 min at 600× *g*, and resuspended in PBS with 2% FBS (Gibco™). The subsequent suspension was mixed with Guava Nexin^®^ reagent (1:1) and incubated for 20 min at room temperature (RT) protected from light. Sample acquisition was performed using Guava^®^ easyCyte™ 5HT flow cytometer (Merck Millipore Corp., Darmstadt, Germany). The data were analyzed using FlowJo X 10.0.7 software (Tree Star, Inc., Ashland, OR, USA).

### 2.4. Proliferation Index

Proliferation levels were determined by the incorporation of BrdU, a synthetic thymidine analogue, using the APC BrdU Flow Kit (552598; BD Pharmingen™, BD Biosciences, San Jose, CA, USA) according to the manufacturer’s instructions. Briefly, BrdU was incubated for 2 h in self-renewal medium, after the last tBHP treatment (day 5). The incorporated BrdU was stained with APC-conjugated anti-BrdU antibody and samples were acquired using the BD LSRFortessa™ Flow Cytometer (BD Biosciences). FlowJo X 10.0.7 software (Tree Star, Inc.) was used for data analysis.

### 2.5. Transfection Assays

Transfections were performed using Lipofectamine™ 3000 (Invitrogen™, Thermo Fisher Scientific Inc.), according to the manufacturer’s instructions, in both tBHP-treated and untreated NSCs. SIRT3 overexpression was induced at day 2 post-plating, i.e., 4 h after the second exposure to tBHP or vehicle (control), by transfecting NSCs with pCMV6-AC-GFP as control, and the same plasmid encoding human GFP tagged SIRT3 (RG217770). Both plasmids were kindly provided by Dr. Ana Cristina Rego, University of Coimbra. For transfection, Opti-MEM^®^ (Gibco™) containing the mixture of Lipofectamine and DNA (2 µL:1 µg) was added and the NSCs were incubated (37 °C, 5% CO_2_) to the following day until the end of the third tBHP-treatment. To assess the effect of SIRT3 overexpression and LCAD downregulation (co-modulation studies), NSCs were simultaneously co-transfected with SIRT3 or control (the abovementioned plasmids) and with 60 nM of small interfering RNA (siRNA) specific to LCAD (Acadl Silencer Pre-designed siRNA, ID162072, Ambion™, Thermo Fisher Scientific Inc.) or the respective negative control (Silencer^®^ Select Negative Control #1 siRNA, Ambion™). Co-transfection was performed with the same timing as mentioned above, i.e., at day 2 of culturing and after a 4 h resting period post-tBHP incubation). For all transfection assays cells were collected two days afterwards (day 5, at the end of chronic aging treatment) and processed for flow cytometry, immunoblotting, and qRT-PCR.

To evaluate transfection efficiency, both overexpression and silencing, SIRT3 and LCAD levels were determined by Western blot, respectively.

### 2.6. Mitochondrial ROS Detection

mtROS levels were measured using MitoSOX™ Red mitochondrial superoxide indicator (M36008; Invitrogen™). Briefly, NSCs were incubated for 10 min at 37 °C with 5 μM MitoSOX™ Red in Hank’s balanced salt solution (14025; Gibco™). Cells were then washed, collected, and resuspended in DPBS with 2% FBS. Samples were subsequently acquired with Accuri C6 Flow Cytometer (BD Biosciences) and data analysis was performed in FlowJo X 10.0.7 software (Tree Star, Inc.).

### 2.7. ATP Measurement

ATP content was assessed by performing the Mitochondrial ToxGlo™ assay (G8001; Promega Co., Madison, WI, USA), following the manufacturer’s instructions. ATP Detection Reagent was added to NSCs, resulting in cell lysis and the generation of a luminescent signal proportional to the amount of ATP present. Emission of luminescence was detected using the GloMax^®^ 96 Microplate Luminometer (Promega Co.).

### 2.8. Total Protein Extraction

NSCs were collected and lysed using ice-cold lysis buffer (10 mM Tris-HCl, pH 7.6, 5 mM MgCl_2_, 1.5 mM potassium acetate, 1% Nonidet P-40, 2 mM dithiothreitol and 1X Halt Protease and Phosphatase Inhibitor Cocktail (Thermo Fisher Scientific Inc.)) for 30 min in ice. Samples were subsequently sonicated for 30 s and centrifuged for 10 min at 3200× *g* at 4 °C. The supernatant was recovered and stored at −80 °C. Protein content was measured by the Bio-Rad protein assay kit (5000002; Bio-Rad Laboratories, Hercules, CA, USA), according to the manufacturer’s specifications, using bovine serum albumin as standard.

### 2.9. Immunoblotting

Protein levels of p16, p21, p53, NeuN, Sox2, SIRT3, LCAD, acetyl-SOD2, and SOD2 were determined by Western blot analysis. Briefly, 40 µg of protein extracts were separated on a 12% sodium dodecyl sulfate-polyacrylamide gel electrophoresis and then transferred onto a nitrocellulose membrane and blocked with 5% milk solution. Uniform protein loading and transfer was confirmed by transient staining with 0.2% Ponceau S (Sigma-Aldrich Corp.). Blots were incubated overnight with mouse primary antibodies reactive to p53 (1:200; sc-99; Santa Cruz Biotechnology Inc., Dallas, TX, USA), NeuN (1:500; MAB377; Merck Millipore) or rabbit primary antibodies reactive to p16 (1:200; sc-1207; Santa Cruz Biotechnology Inc.), p21 (1:200; sc-397; Santa Cruz Biotechnology Inc.), Sox2 (1:500; AB5603; Merck Millipore), SIRT3 (1:1000; D22A3; Cell Signaling Technology, Danvers, MA, USA), LCAD (1:1000; 17526-1-AP; Proteintech, Rosemont, IL, USA), acetyl-SOD2 (1:1000; acetyl-K68; ab137037; Abcam, Cambridge, UK), and SOD2 (1:200; sc-30080; Santa Cruz Biotechnology Inc.) Blots were subsequently incubated with anti-mouse or anti-rabbit secondary antibodies conjugated with horseradish peroxidase (1:5000, Bio-Rad Laboratories) for 2 h at RT. Membranes were processed for protein detection using the Immobilon™ Western Chemiluminescence HRP Substrate (WBKLS0500; EMD Millipore, Merck Millipore Corp.) in a ChemiDoc MP system (Bio-Rad Laboratories). Finally, the relative intensities of protein bands were analyzed using the ImageLab Version 5.1 densitometric analysis program (Bio-Rad Laboratories).

### 2.10. Immnunoprecipitation Assay

The physical association of SIRT3 and LCAD was detected by immunoprecipitation analysis. In brief, whole cell extracts were prepared by lysing cells in lysis buffer (50 mM Tris-HCl pH 7.4, 180 mM NaCl, 1 mM EDTA, 0.5% Triton X-100, and 1X Halt Protease and Phosphatase Inhibitor Cocktail). Immunoprecipitation experiments were carried out using the antibody specific to LCAD (17526-1-AP; Proteintech) and Ezview Red Protein G Affinity Gel (Sigma-Aldrich Corp.). Typically, 500 μg of lysate was incubated overnight with 1 μg of the abovementioned primary rabbit polyclonal antibody to LCAD at 4 °C. Immunoblots were then probed with LCAD antibody.

SIRT3 (1:1000; D22A3; Cell Signaling Technology) and acetyl-lysine (1:1000; ab190479; Abcam) expression were determined in the same membrane after stripping off the immune complex for the detection of LCAD. In parallel, immunoprecipitation assays using rabbit monoclonal antibodies reactive to IgG were used as controls. The results of SIRT3 and acetyl-lysine after LCAD immunoprecipitation were normalized with those obtained using rabbit monoclonal antibodies reactive to IgG immunoprecipitation assays as well as with the LCAD total levels. The initial input was also used as a loading control of LCAD in immunoprecipitation analysis (Appendix A).

### 2.11. Total RNA Extraction and Quantitative RT-PCR (qRT-PCR)

Total RNA was extracted using 0.5 mL TRIzol^®^ reagent (15596; Invitrogen™) per sample. After mixing with 0.1 mL chloroform and centrifuging at 12,000× *g* for 15 min at 4 °C, the aqueous phase was collected and total RNA precipitated by incubation with 0.25 mL isopropyl alcohol at −20 °C for 1 h. Samples were centrifuged at 12,000× *g* for 10 min at 4 °C and the RNA pellet was washed with 75% ethanol and centrifuged at 7500× *g* for 5 min at 4 °C. RNA pellets were air dried and resuspended in RNase-free water. The purity of the RNA was checked and DNA contaminations were eliminated with DNase I recombinant (04716728001; Roche Applied Science, Mannheim, Germany) following the manufacturer’s instructions.

cDNA was prepared from 1000 ng total RNA using NZY Reverse Transcriptase (MB12402; NZYTech, Lisbon, Portugal) according to manufacturer’s instructions. Real-time RT-PCR was performed using a SensiFastTM SYBR^®^ Hi-ROX kit (BIO-92020; Bioline USA Inc., Taunton, MA, USA) in an Applied Biosystems QuantStudio 7 Flex Real-Time PCR system (Thermo Fisher Scientific Inc.). Primer sequences are listed in Table 1. Relative gene expression was calculated based on the standard curve and normalized to the level of hypoxanthine phosphoribosyltransferase (*Hprt*) housekeeping gene and expressed as fold change from controls.

### 2.12. Senescence Associated β-Galactosidase Activity

Senescence was determined by staining cells for SA-β-galactosidase detection, that is, a widely used biomarker of cellular senescence [43]. Cells were treated with tBHP for 4 days and co-transfected for SIRT3 overexpression and/or LCAD silencing at day 2 post-plating (as described in Section 2.5). Subsequently, 24 h before the last tBHP treatment (or vehicle), a senescence staining protocol was performed using specific commercial kits, according to the manufacturer’s instructions. More specifically, (a) for the tBHP validation model, the Cellular Senescence Assay kit (KAA002RF; Merck Millipore) and (b) for co-transfections, the Senescence β-galactosidase Staining kit (9860; Cell Signaling Technology). Regarding the co-transfections assays, for the positive control of senescence the induction of oxidative stress was performed by an overnight incubation with 25 µM H_2_O_2_. Images were acquired for three independent experiments (in each one, all the six groups were run simultaneously) with the Invitrogen™ EVOS FL Auto 2 Imaging system, model AMAFD2000. At least seven images were obtained for each group condition, and on average more than 250 total cells were scored for tBHP treatment groups. Scoring was performed with ImageJ Fiji 1.53c Software. The total cell population and SA-β-gal positive cells (blue color) were counted to calculate the percentage of SA-β-gal positive cells for a given group.

### 2.13. Animal Model of Depressive-Like Behavior and Physical Exercise (PE) Protocol

Adult male C57BL/6 mice (20 ± 2 g, 8–12 weeks old; Charles River Laboratories, Barcelona, Spain) were group-housed and kept under standard laboratory conditions (22 ± 2 °C; 12 h light/12 h dark cycles; ad libitum access to food and water), before exposure to an unpredictable chronic mild stress (uCMS) regimen to induce depressive-like behavior.

One week after acclimatization to iMM’s rodent facility, mice were subjected to different mild stressors in a random and unpredictable fashion several times a day for 8 weeks (uCMS group). The selected stressors included reversed light/dark cycles, confinement to a restricted space (1–3 h), cage shaking (1–3 h), strobe illumination (1–3 h) and overnight housing on damp bedding or placement in a tilted cage (40°) with restricted access to food and water. Unstressed mice (control group) were handled only for cage changes. Following the uCMS induction regimen, one group of uCMS mice was exposed to a PE protocol over a 2-week period with a concomitant mild uCMS as described elsewhere [44,45]. Briefly, mice were exposed to 30 min daily sessions on a treadmill (LE8710MTS, Panlab/Harvard Apparatus S.L.U.) at a 10 m/min speed in the first week and at a 15 m/min speed in the second week (uCMS + PE group). During this time, the other uCMS mice and control group were handled without PE.

At the end of experiments, all mice were anesthetized with isoflurane and perfused transcardially with PBS. After decapitation, their brains were carefully removed and dissected into the two cerebral hemispheres on a cold platform. The SVZ neurogenic region was isolated from one hemisphere and flash-frozen in liquid nitrogen for mRNA analysis as aforementioned (see Section 2.11). The contralateral hemisphere was post-fixed in 4% paraformaldehyde (24 h, 4 °C) for immunohistochemistry assays.

### 2.14. Free-Floating Immunohistochemistry

After fixation, brain hemispheres were cryoprotected with a 30% sucrose solution (48 h, 4 °C), then gelatin-embedded and coronally sectioned (40 µm thickness), from the olfactory bulb to the hippocampus (Leica CM 3050S cryostat; Leica Biosystems, Wetzlar, Germany), to produce free-floating sections that were collected into 10 series. Each series contained an anterior–posterior reconstruction of brain slices separated by 400 μm between them. Free-floating sections were stored at −20 °C in anti-freezing medium, and only the ones located at the level of SVZ were processed for immunohistochemistry.

Tissue sections were degelatinized in PBS at 37 °C and then subjected to antigen retrieval in 10 mM sodium citrate buffer, pH 6.0, for 25 min at 80 °C. After repeated washing in PBS (3 × 10 min at RT), nonspecific staining was blocked for 1 h at RT with blocking solution (1% BSA, 2% Triton-X-100, 10% normal donkey serum in PBS). Subsequently, slices were incubated for 2 days at 4 °C with both primary antibodies goat anti-doublecortin (DCX) (1:500; sc-8066; Santa Cruz Biotechnology Inc.) and rabbit anti-SIRT3 (1:100; D22A3; Cell Signaling Technology) diluted in blocking solution. After rinsing in PBS, a mixture of the secondary antibodies anti-goat Alexa Fluor 488 and anti-rabbit Alexa Fluor 568 (all 1:500; Thermo Fisher Scientific Inc.) plus the DNA stain Hoechst 33258 (5 µg/mL; Sigma-Aldrich Corp.) was incubated for 2 h at RT. Finally, tissue samples were mounted on Superfrost Plus™ microscope slides (Thermo Scientific) using Mowiol 4-88 (Calbiochem, San Diego, CA, USA). Negative controls, in which the primary antibodies were omitted, were performed simultaneously to validate the results.

All samples were analyzed in a Zeiss Cell Observer SD spinning disk confocal microscope using ZEN 2.6 Software (Carl Zeiss Inc., Jena, Germany). Immunofluorescence SVZ images were captured with a 63x Oil Immersion Plan-Apochromat DIC (NA 1.4) objective (representative images) and a 40x Water Immersion LD C-Apochromat (NA 1.1) objective (for quantification analysis). All acquisition conditions were kept constant between samples during the capture process. Tissue background autofluorescence was determined and subtracted for immunofluorescence quantification. Maximum intensity Z-stack projection was carried out and ImageJ Fiji 1.53c software was used to measure the intensities of fluorescence signals for DCX (green) and SIRT3 (red), after grayscale threshold determination. Using Hoechst and DCX staining, a defined area in the SVZ region was manually traced (ROI) and was considered in all images for fluorescence quantification. A total of eight brain slices per group condition were evaluated (one or two measurements per slice, four mice per group) rendering between 10 to 13 data points per group. The researcher performing image analysis was blind to the experimental group to which the animals belonged.

### 2.15. Statistical Analysis

Statistical significance was assessed using an unpaired two-tailed Student’s t test when two groups where compared, and a one-way ANOVA followed by Dunnett’s post-test for multiple comparisons to one control or a two-way ANOVA followed by the Tukey post-test for multiple comparisons. All statistical analyses were performed using GraphPad Prism 6.01 (GraphPad Software, Inc., San Diego, CA, USA). Values of *p* < 0.05 were considered significant.

## 3. Results

### 3.1. Tert-Butyl Hydroperoxide Induces Aging Features in NSCs and Reduces Mitochondrial Antioxidant Defence System

Adult neurogenesis is strongly impaired during aging and psychiatric disorders, including anxiety and depression [14,15]. Although a healthy balanced diet and a regular exercise practice have been extensively described as key factors to arrest aging and mood disorders [46,47], the precise role of metabolism in abrogating NSC dysfunction under these two scenarios has never been reported. To further dissect the involvement of mitochondrial players in abrogating age-related NSC changes, we first used a cell line established using a method that produces pure cultures of adherent NSCs. These cells, continuously expanding by symmetrical division and capable of tripotential differentiation [39,41], were incubated with small levels of the prooxidant molecule tert-butyl hydroperoxide (tBHP) for 4 days, as described in Section 2. Our results show that chronic treatment with tBHP induced alterations in the fate of NSCs, diminishing NSC viability (Figure 1A, left panel), but, further, their proliferation potential, as assessed by BrdU incorporation (*p* < 0.001) (Figure 1B). In fact, the decrease in NSC proliferation after tBHP exposure was accompanied with the presence of senescence signals in NSCs, such as β-galactosidase droplets (Figure 1A, center panel) and high levels of the cell-cycle-arrest proteins p16, p21, and p53 (*p* < 0.001) (Figure 1A, right panel). We also found that, along with proliferation deficits, the differentiation potential of NSCs was also markedly affected after tBHP incubation, since immunoblot analysis showed a significant decrease in neural stem cell (Sox2) and neuronal (NeuN) markers (*p* < 0.05 and *p* < 0.001, respectively) (Figure 1C). More interesting, SIRT3, a central player in mitochondrial metabolism and oxidative protection, was also markedly reduced in this cellular context (*p* < 0.01) (Figure 1D). These results demonstrate that tBHP induces a feasible NSC aging model, possibly through a reduction in the SIRT3-dependent mitochondrial antioxidant defense system.

### 3.2. SIRT3 Rescues Mitochondrial Oxidative Stress and Differentiation Potential of Aged NSCs

To further investigate the role of SIRT3 in regulating NSC behavior under aging and in an oxidative context, we overexpressed SIRT3 in tBHP-induced NSC aging and re-evaluated NSC proliferation and the differentiation potential of SIRT3 overexpressing aged NSCs. After 24 h of transfection, SIRT3 protein levels significantly increased, when compared with control transfected cells (Figure 2A). Interestingly, qRT-PCR experiments showed that increased levels of SIRT3 are only capable of rescuing the differentiation, not the proliferation potential of NSCs (Figure 2B). In fact, SIRT3 overexpression in aged NSCs (Figure 2A) markedly increased the mRNA levels of the neuronal marker MAP2 (*p* < 0.05) but had no effect on the proliferation marker ki67.

SIRT3 is the main positive regulator of the superoxide dismutase 2 (SOD2), the most known scavenger of mitochondrial reactive oxygen species (ROS) in cells. The SIRT3-induced SOD2 activation results from SOD2 deacetylation promoted by SIRT3 [32]. Therefore, we evaluated the protein levels of acetyl-SOD2, SOD2, and mitochondrial ROS (mtROS) in aged NSCs, and particularly, the role of SIRT3 in abrogating mitochondrial oxidative stress in this cellular context. Our results revealed that although the chronic oxidative insult with tBHP did not alter the total levels of acetyl-SOD2 and SOD2 content (Figure 2C), it triggered a significant increase in mtROS in NSCs (*p* < 0.01) (Figure 2D). Conversely, tBHP strongly reduced cellular ATP supplies (*p* < 0.01) (Figure 2E). Upregulation of SIRT3, in turn, restored ATP production (Figure 2E) and significantly reduced the ratio of acetyl-SOD2/SOD2 levels (*p* < 0.001) (Figure 2C) decreasing mtROS in aged NSCs (*p* < 0.05) (Figure 2D). Hence, these results demonstrate that, under oxidative conditions, SIRT3 directs NSCs towards differentiation while also abrogating mitochondrial oxidative damage and increasing the bioenergetic levels of these cells.

### 3.3. SIRT3 Further Activates the Long Chain Acyl-CoA Dehydrogenase (LCAD) in Aged NSCs

It is well-established that the differentiation and activation of NSCs is an ROS-dependent process, in which mtROS act as key cellular signals to block NSC proliferation [48]. On the other hand, SIRT3 has other important metabolic targets, such as LCAD, an enzyme involved in mitochondrial β-oxidation of unsaturated fatty acid lipid metabolism [36]. Since we recently demonstrated that LCAD levels are increased throughout the differentiation of NSCs, we hypothesized that SIRT3 abrogates age-related neurogenesis impairment, by activating LCAD. To clarify this hypothesis, we investigated the physical interaction between SIRT3 and LCAD in NSCs undergoing age-induced alterations. Indeed, immunoprecipitation assays demonstrated that SIRT3 physically interacts with LCAD in NSCs but this interaction is diminished under tBHP-induced aging conditions (*p* < 0.001) (Figure 3A). Notably, upregulation of SIRT3 in aged NSCs re-established the interaction levels between SIRT3 and LCAD (Figure 3A). To clarify whether SIRT3 and LCAD interaction would result in LCAD activation, we then evaluated the levels of LCAD acetylation in aged NSCs overexpressing SIRT3 (Figure 3B). Interestingly, immunoprecipitation experiments showed that the inactive form of LCAD, assessed by its acetylation levels, was significantly increased in NSCs undergoing tBHP-induced aging conditions (*p* < 0.001) (Figure 3B). In contrast, in SIRT3 overexpressing NSCs (with or without chronic tBHP treatment), the levels of LCAD acetylation were similar to those found in untreated NSCs. These results demonstrated that, in NSCs, SIRT3 physically interacts with LCAD to promote its activation, further inducing its activation in aging conditions.

### 3.4. SIRT3 Requires LCAD and Oxidative Control to Decelerate NSC Aging

To better understand the mechanism by which SIRT3 modulates the fate of NSCs throughout aging, namely whether LCAD activation and the balance of mtROS were linked to previously observed SIRT3-rescued effects, we performed co-modulation assays in this cellular context. Therefore, the expression levels of the neuronal marker MAP2 were re-evaluated in NSCs co-transfected with SIRT3 overexpression plasmid and siRNA LCAD. Interestingly, the knockdown of LCAD revealed that SIRT3 requires this lipid metabolism enzyme to prevent age-induced impaired neurogenesis (Figure 4A). To further understand the role of general lipid metabolism on SIRT3-mediated effects, we also evaluated the expression levels of a cytosolic lipid droplet binding protein, Perilipin 2 (PLIN2), in this NSC context. Interestingly, our results showed that PLIN2 expression significantly decreased in aged NSCs, being only prevented by SIRT3 when LCAD was not silenced. These data reinforce the role of lipid metabolism in SIRT3-mediated effects on NSCs (Figure 4A). Regarding the effects of tBHP in inducing NSC senescence, these were re-evaluated in both NSCs co-transfected with SIRT3 overexpression plasmid and LCAD siRNA, as well as in SIRT3 overexpressing NSCs treated with a prooxidant molecule, hydrogen peroxide (H_2_O_2_) (Figure 4B). In fact, senescence-associated growth arrest has been shown to deeply dependent on oxidative stress-mediated regulation [49]. Our results showed tBHP-induced NSC senescence was abrogated in NSCs by the upregulation of SIRT3 (*p* < 0.001) (Figure 4C). Nevertheless, the SIRT3 protective effect was reverted when LCAD was simultaneously downregulated and/or a prooxidant cell environment was created (*p* < 0.001 vs. SIRT3 overexpressing cells with or without tBHP) (Figure 4C). These results strongly support the idea that SIRT3 requires to control both oxidative state and lipid metabolism of long chain fatty acids to rescue aged-induced NSC senescence and impairment of neurogenesis.

### 3.5. Reduced Neurogenesis in Depressive-Like Mice Is Associated with LCAD Downregulation

Finally, the SIRT3 regulatory network was investigated in vivo using the unpredictable chronic mild stress (uCMS) paradigm to mimic depressive-like behavior in mice. Adult male mice were chronically exposed to unpredictable mild stressors over 8 weeks, as indicated in Section 2. Controls were age-matched mice without stress exposure. After an initial 8-week period of uCMS, half of the animals in the uCMS group were subjected to a physical exercise (PE) protocol for 2 weeks. This procedure aimed to clarify whether PE, a well-established inducer of SIRT3 [50], could have any impact on an impaired neurogenesis-associated pathology, such as depression. Animals were tested on a battery of behavioral and memory tasks to validate the depression phenotype of these mice. After 8 weeks of mild stress, all animals subjected to the uCMS protocol had developed depressive behavior. uCMS animals that underwent PE treatment (uCMS+PE) showed a partial recovery of depressive-like symptoms (data not shown—manuscript in preparation). More importantly, immunohistochemistry for DCX, a marker of new-born neurons, showed that the SVZ of uCMS mice presented lower levels of fluorescence intensity for DCX when compared with unstressed controls (*p* < 0.001) (Figure 5A,B). Notably, PE was capable of rescuing neurogenesis levels in stressed animals (*p* < 0.01) (Figure 5A,B). To clarify the involvement of SIRT3 in regulating neurogenesis in these mice, we evaluated the SIRT3 levels in the same neurogenic SVZ regions and, surprisingly, a significant increase in SIRT3 levels was observed in uCMS mice, when compared with the control and uCMS-PE groups (*p* < 0.001) (Figure 5B). To further understand the inconsistent data regarding SIRT3 and neurogenesis levels in this model of depressive-like behavior, we assessed the expression levels of several mitochondrial regulators, including LCAD but also players of mitochondria biogenesis, such as the peroxisome proliferator-activated receptor γ coactivator α (PGC-1α) and the mitochondrial transcription factor A (Tfam). In fact, although uCMS mice presented evident amounts of SIRT3, our results revealed that LCAD expression was significantly compromised in these animals when compared with the control group (*p* < 0.01) (Figure 5C). Interestingly, in agreement with what we found for DCX^+^ cells, the PE was capable of restoring LCAD levels in uCMS mice (*p* < 0.05). The expression levels of PGC-1α were not affected by PE in uCMS mice, and although we found a marked decrease in Tfam, PE did not affect the expression levels of this mitochondrial transcription factor in these specific neurogenic niches. Thus, targeting NSC metabolism, namely through SIRT3-mediated LCAD activation, appears to be a suitable strategy to recover adult neurogenesis and confer stress resilience.

## 4. Discussion

The world’s population is aging but most people live their lives in a reactive health response system, only seeking treatments when they have specific symptoms. Therefore, the identification of effective and non-pharmacologic strategies capable of improving brain homeostasis and physiological neural repair in asymptomatic adulthood represents a right step to delay the new cases of neurodegeneration and psychiatric disorders identified every year. Adult NSCs are pivotal to reinforce the adult synaptic network [51], assure cognitive performance, and promote resilience to chronic stress [15,52]. However, the number and activity of these cells drop throughout life. The poor survival and differentiation levels of NSCs have been one of the major drawbacks of these cells. It has been shown that lifespan of NSC activity, including their proliferation and differentiation potential, is deeply dependent on their metabolic changes. Here, we uncovered a new role of SIRT3, a major regulator of mitochondrial metabolism, in mediating the fate of NSCs under conditions of aging and psychiatric stress. Our data clearly demonstrate that activation of lipid metabolism by SIRT3 is essential for both the preclusion of NSC senescence and proper NSC differentiation.

Throughout life, adult neurogenic niches experience a progressive decline in homeostatic and regenerative abilities, greatly attributed to redox cues of age-associated intracellular and extracellular changes [53]. In the present study, we began to validate an in vitro model of NSC aging induced by tert-butyl hydroperoxide, tBHP. Indeed, the chronic treatment of NSCs with this prooxidant molecule resulted in features of cell aging, including senescence and cell cycle arrest. In accordance with previous reports, the proliferation and differentiation potential of NSCs were also found to be significantly reduced by tBHP, corroborating the fact that this molecule is indeed capable of inducing a feasible aging model in NSCs [54]. More importantly, tBHP-induced NSC aging was associated with a reduction of the mitochondrial regulator SIRT3 levels. Although it is possible that other SIRTs, such as SIRT1 and SIRT2, might also regulate neurogenesis, aging, and even depression, we decided to focus only on SIRT3 in this study, given its central role in mitochondrial metabolism and oxidative protection. In fact, the individual metabolism slows down during aging, rendering the aging brain more vulnerable to oxidative and cell damage. These compromised bioenergetics during aging result, in turn, in a wide range of downstream alterations, including inflammation. After dissecting the underlying mechanism of SIRT3 in NSCs, we found that age-reduced SIRT3 was responsible for triggering senescence and the differentiation decline of NSCs. In fact, forced SIRT3 upregulation recovered the expression levels of the neurogenic marker MAP2 and abrogated senescence signals of NSCs. The effect of SIRT3 in rescuing NSC differential potential could be further explored in the future, namely by the evaluation of additional neuronal and glial markers, since the precise role of SIRT3 in NSC lineage determination is still not clear. Unexpectedly, the proliferation rate, which may be regulated by SIRT3-independent pathways, was not influenced. We also showed that increased levels of SIRT3 in NSCs resulted in the activation of the major superoxide-scavenger in mitochondria, SOD2, and consequently in a marked reduction of mtROS. However, based on the role of both ROS and LCAD in the neural differentiation process [23], and the fact that LCAD is one of other major targets of SIRT3 in cells [36], we hypothesized that LCAD could be a major player in SIRT3-mediated NSC differentiation. We first confirmed the physical interaction between SIRT3 and LCAD in our cells. Our results showed for the first time that SIRT3 physically interacts with LCAD in NSCs, promoting its activation, particularly in the context of aging. More importantly, co-modulation of SIRT3 and LCAD confirmed our hypothesis that this enzyme of lipid degradation is indeed required by SIRT3 to restore the balance of lipid storage-related proteins, such as PLIN2, and rescue the neurogenic potential of NSCs.

In age-related diseases, cell senescence has been shown to be modulated by both redox and metabolic shifts [55,56]. Here, we also revealed that SIRT3-inhibited NSC senescence really depends on SIRT3’s role in mediating both LCAD activation and oxidative protection. Thus, it is tempting to think that the reinforcement of both oxidative protection and metabolic rate are perfectly orchestrated by SIRT3 to mitigate NSC senescence. In fact, this idea is also in agreement with a growing body of literature, in other cell types, indicating that SIRT3 counteracts senescence by targeting SOD2 [49,57]. In this regard, it would be also interesting to evaluate changes in SOD2 activation in LCAD downregulated aged NSCs.

The lifelong activity of adult NSCs is markedly dependent on individuals’ age but also on different components of their environment and lifestyles, including diet and physical exercise (PE) [46,47]. On the other hand, adult neurogenesis was shown to buffer stress responses and depressive behaviors [15]. Therefore, we decided to also explore the regulatory network of SIRT3-induced neurogenesis in an in vivo model of depressive-like behavior. Evidence supports a role for PE in delaying aging and increasing stress resilience [58], while several studies have already demonstrated an upregulation of SIRT3 activity in neurons by exercise [50]. However, no information has yet been reported for the role of SIRT3 and PE in mediating stress resilience and cognition, namely through NSC activity. For this reason, a group of animals subjected to chronic stress undergoing PE was also included in our study. Interestingly, stressed mice presented significantly lower levels of neurogenesis in SVZ, which was partially rescued by PE. The biological link between adult neurogenesis and depression has been already elucidated. In fact, several studies have demonstrated that human adult neurogenesis is severely depleted in major depressive disorder, as indicated by the reduction of distinct neurogenic markers and neurogenic niche volumes [14,59]. Further, the observation that the treatment of depression by antidepressants depends on an increased rate of neurogenesis also corroborates the key role of adult neurogenesis on this type of neurological disorder [59]. Although the majority of studies reporting a role for adult neurogenesis and stress resilience have been performed in hippocampal neurogenic niches, it has been already shown that the adult SVZ neurogenic process is also compromised under unpredicted prolonged stress [60,61]. Curiously, it has been recently demonstrated that both anxiety- and depression-like states lead to marked olfactory deficits along with impaired adult neurogenesis [18]. Indeed, the SVZ-derived neural progenitors migrate towards the olfactory bulb, where they differentiate into local neurons involved in olfactory system [62], while an association between olfactory function and depression has also been revealed [63]. In this regard, it was recently proposed that the rejuvenation of SVZ neurogenesis would be a promising strategy to arrest aging and come to resemble a youthful and health brain [64]. Certainly, more studies will help us to further clarify the precise role of SVZ neurogenesis in abrogating stress-induced depression behavior.

Notably and unexpectedly, depressive mice (uCMS) presented high levels of SIRT3 when compared with control and stressed mice undergoing physical exercise (uCMS+PE). We might speculate that the increase in total SIRT3 levels observed in the brains of uCMS mice could represent an adaptative cellular mechanism to counteract the decrease in NSC activation under a harmful scenario, not necessarily correlated with SIRT3 activation levels. More importantly, although depressive mice present high levels of SIRT3, the LCAD expression was found significantly reduced in the neurogenic niches of these mice. In fact, recent discoveries have also revealed that mental illness and obesity are common conditions that tend to co-occur within individuals [65]. Besides the absent effect of PE on mitochondrial-biogenesis-related markers, the lipid metabolism of unsaturated fatty acids was completely restored by PE in uCMS mice, possibly explaining the re-establishment of neurogenesis observed in the same animals. This observation is in line with previous studies showing the positive effect of PE in increasing fatty acid oxidation and lipid metabolism [66]. However, exploring possible changes in lipidomics upon PE-, depression- and SIRT3-mediated effects appears to be of upmost importance for future studies.

Altogether, these findings led us to hypothesize that exercise training increases the lipid metabolic rate, and that, under stress- and depression-like scenarios, SIRT3 cooperates with PE-induced LCAD to allow a proper neuronal differentiation process. However, further experiments will be required to prove this hypothesis in vivo.

Indeed, it has become clear that certain factors of environmental enrichment, such as PE, require adult neurogenesis to facilitate the recovery from psychosocial stress [47]. Although the mechanisms by which PE protects the brain from stress-induced depression would involve a wide range of molecular mechanisms, including tryptophan’s metabolites [67], the intrinsic metabolic changes in PE responsible for lifelong adult neurogenesis are not completely understood. They will be surely crucial to arrest aging and stress-induced depression, since exercise has been shown to increase the NSC pool and neurogenesis in adult mice [46].

The pivotal role of mitochondrial switching in regulating the fate of NSC has been already well-established and -characterized by our group and others [19,68,69,70,71]. In past years, we demonstrated that a strong inhibitor of mitochondrial apoptosis and a potent neuroprotective molecule in animal models of neurodegenerative diseases [72] increases the NSC pool and neurogenesis in adult rats by preventing mitochondrial dysfunction and increasing ATP levels [69,70]. More recently, we also identified a fatty acid metabolic reprograming in NSCs treated with such molecule [23]. Interestingly, we recently demonstrated that a specific high caloric diet is capable of upregulating microbial metabolisms that, in turn, induce premature neuronal differentiation and depletion of NSC pool in adult mice in a mitochondrial and oxidative stress-dependent manner [71]. Indeed, mice fed with a high fat diet have been shown to present anxiety- and depression-like behavior [73]. In agreement with this, this type of diet has also been reported to induce memory impairment, depressive-like behavior, and stress-induced depression by inhibiting adult neurogenesis [74], further demonstrating the strong link between host metabolism, adult neurogenesis, and mental health.

Despite the well-established link between mitochondrial metabolism and neurogenesis, the precise molecular changes underlying NSC differentiation in aging- and stress-induced depression scenarios have never been explored. Herein, we dissected the mechanism by which a master mitochondrial regulator, SIRT3, regulates the fate of NSC in these contexts. We clearly demonstrated that, under age-related oxidative conditions, SIRT3 directly interacts and activates a key enzyme of lipid metabolism, LCAD, to inhibit NSC senescence and enable NSC differentiation. We also showed that exercise training restores the levels of LCAD in a mouse model of stress-induced depression, reestablishing functional neurogenesis.

## Figures and Tables

**Figure 1 cells-11-00090-f001:**
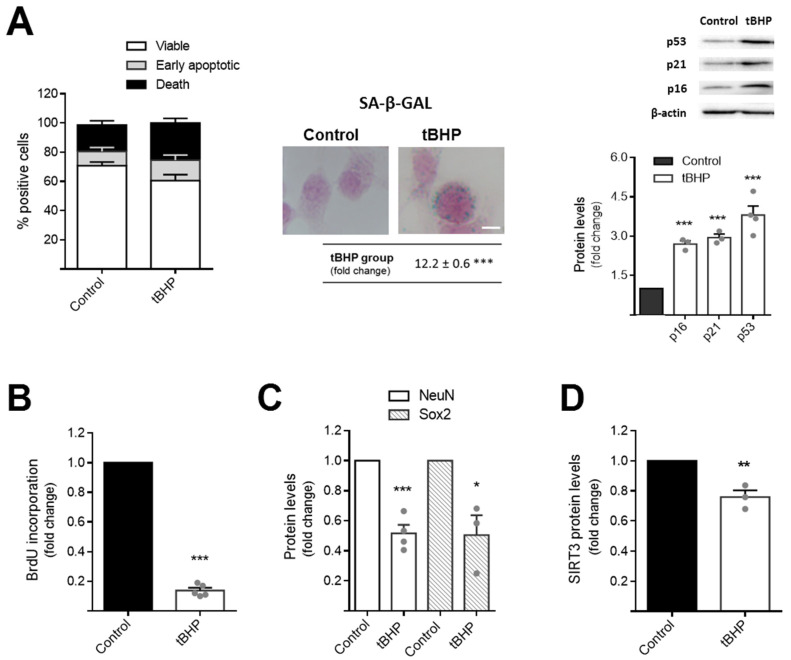
tBHP induces aging features in NSCs and reduces SIRT3 levels. Mouse NSCs were treated with 50 μM tBHP for short periods of 2 h, for 4 consecutive days, in self-renewal conditions. After the fourth tBHP-treatment, NSCs were collected to assess viability, proliferation, and protein expression as described in Section 2. Senescence staining protocol was performed 24 h before the last tBHP treatment. (**A**) Quantification of NSC viability (left); senescence SA-β-galactosidase detection (center, representative images of bright-field microscopy; scale bar: 10 μm; and quantitative analysis); expression levels of cell-cycle-arrest proteins p16, p21, and p53 (right, immunoblots and densitometry analysis. β-Actin was used as loading control). (**B**) Quantification of proliferation by BrdU incorporation. Immunoblot densitometry analysis reflecting the total protein levels of (**C**) Sox2—stemness marker, NeuN—neuronal marker, and (**D**) the antioxidant mitochondrial protein SIRT3. Data is expressed as fold change over control and represent mean values ± SEM for at least three individual experiments. Each data point represents an individual value. * *p* < 0.05, ** *p* < 0.01, and *** *p* < 0.001 compared to control NSCs. Abbreviations: BrdU, bromodeoxyuridine; NeuN, neuronal nuclei; SA-β-GAL, SA-β-galactosidase; Sox2, SRY (sex determining region Y)-box 2; tBHP, tert-butyl hydroperoxide.

**Figure 2 cells-11-00090-f002:**
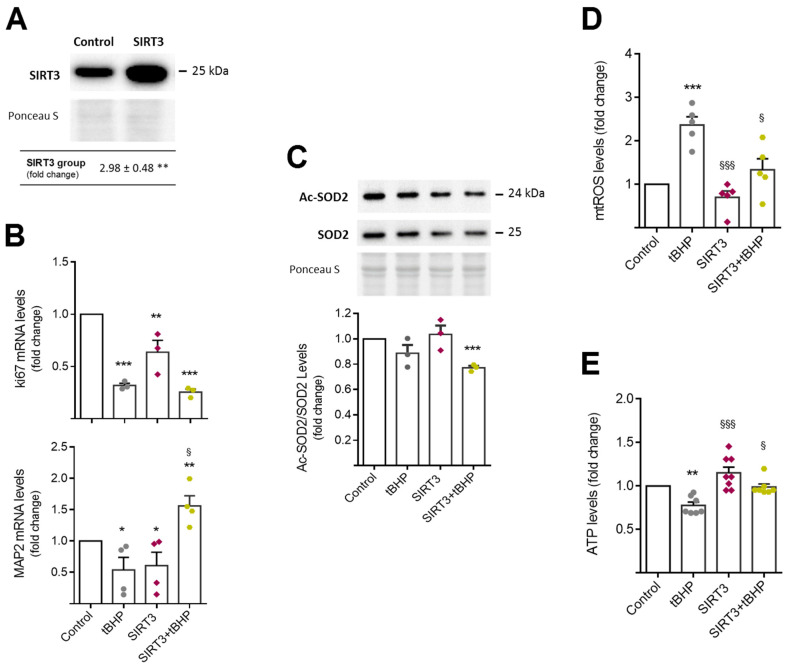
Mitochondrial oxidative damage and differentiation potential of aged NSCs are rescued by SIRT3. Mouse NSCs were exposed to 50 μM tBHP treatment for 4 days (2 h/day), in self-renewal conditions. After the fourth tBHP-treatment, cells were collected for luminescence detection, western blot, qRT-PCR and flow cytometry analysis as described in Section 2. (**A**) Immunoblotting and densitometry of SIRT3 overexpression assessed 24 h post-transfection. (**B**) qRT-PCR analysis of proliferation and differentiation markers, i.e., ki67 and MAP2, in aged NSCs. *Hprt* was used as loading control. (**C**) Immunoblotting (top) of Ac-SOD2 and total SOD2 protein levels, and respective ratio from densitometry analysis (bottom). Quantification of mitochondrial ROS levels (**D**) and ATP levels (**E**). Data is expressed as fold change over control. Data represent mean values ± SEM for at least three individual experiments. Each data point represents an individual value. * *p* < 0.05, ** *p* < 0.01, and *** *p* < 0.001 compared to control cells; ^§^
*p* < 0.05 and ^§§§^
*p* < 0.001 compared to tBHP treated cells. Abbreviations: Ac-SOD2, acetyl-SOD2; MAP2, microtubule-associated protein 2; SOD2, superoxide dismutase 2; tBHP, tert-butyl hydroperoxide.

**Figure 3 cells-11-00090-f003:**
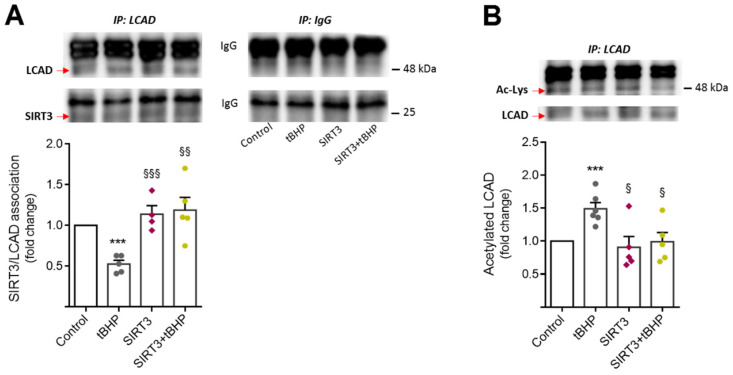
LCAD is further activated by SIRT3 in aged NSCs. Mouse NSCs were exposed to 50 μM tBHP treatment for 4 days (2 h/day), in self-renewal conditions. At day 2 post-plating, cells were transfected with SIRT3 overexpression plasmid or control plasmid. After the fourth tBHP-treatment, total proteins were collected for immunoprecipitation assays as described in Section 2. (**A**) Representative immunoblots with LCAD and SIRT3 specific antibodies (top) and the SIRT3/LCAD association (densitometry analysis, bottom). (**B**) Representative immunoblots with acetyl-lysine and LCAD and specific antibodies (top) and levels of acetylated LCAD (densitometry, bottom). All densitometry values for SIRT3 and acetyl-lysine were normalized to the respective LCAD expression. Data are expressed as fold change over control. Data represent mean values ± SEM mean values for at least four individual experiments. Each data point represents an individual value. *** *p* < 0.001 compared to control cells; ^§^
*p* < 0.05, ^§§^
*p* < 0.01, and ^§§§^
*p* < 0.001 compared to tBHP-treated cells. Abbreviations: Ac-Lys, acetyl-lysine; IP, immunoprecipitation; LCAD, long chain acyl-CoA dehydrogenase; tBHP, tert-butyl hydroperoxide.

**Figure 4 cells-11-00090-f004:**
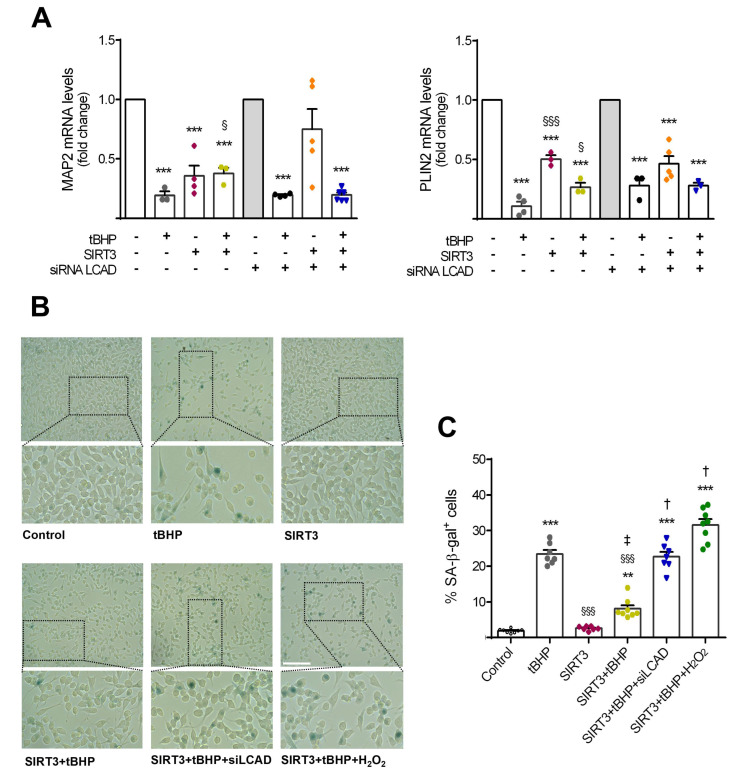
SIRT3 requires LCAD and oxidative control to prevent NSC aging. Mouse NSCs were treated with 50 μM tBHP for 2 h/day, for 4 consecutive days, in self-renewal conditions. At day 2 post-plating, cells were co-transfected with SIRT3 overexpression plasmid and siRNA LCAD, and 48 h afterwards cells were collected for analysis, as described in Section 2. Senescence staining protocol was performed 24 h before the last treatment. (**A**) qRT-PCR analysis of differentiation marker MAP2 and lipid accumulation marker PLIN2. *Hprt* was used as loading control. Data are expressed as fold change over control or siRNA LCAD groups (reference conditions, no tBHP added). (**B**) Representative images of SA-β-gal staining in NSCs exposed to tBHP treatment for 4 days (control group, no tBHP added), and subjected to SIRT3 overexpression, with or without additional LCAD silencing. H_2_O_2_ overnight treatment served as a positive control. Scale bar: 100 μm. Lower panels: selected sections enlarged 4×. (**C**) Quantitative analysis of cells positive for SA-β-gal for a given group, expressed as the percentage of the total number of cells. (**A**,**C**) Data represent mean values ± SEM for three independent experiments, yielding at least 7 data points per group. Each data point represents an individual value. ** *p* < 0.01 and *** *p* < 0.001 compared to control cells, ^§^ *p* < 0.05 and ^§§§^ *p* < 0.001 compared to tBHP-treated cells, ^‡^ *p* < 0.05 compared to SIRT3-transfected cells, and ^†^ *p* < 0.001 compared to SIRT3-transfected cells with or without tBHP treatment. Abbreviations: LCAD, long chain acyl-CoA dehydrogenase; MAP2, microtubule-associated protein 2; PLIN2, perilipin 2; SA-β-GAL, SA-β-galactosidase; siLCAD, LCAD silencing; tBHP, tert-butyl hydroperoxide.

**Figure 5 cells-11-00090-f005:**
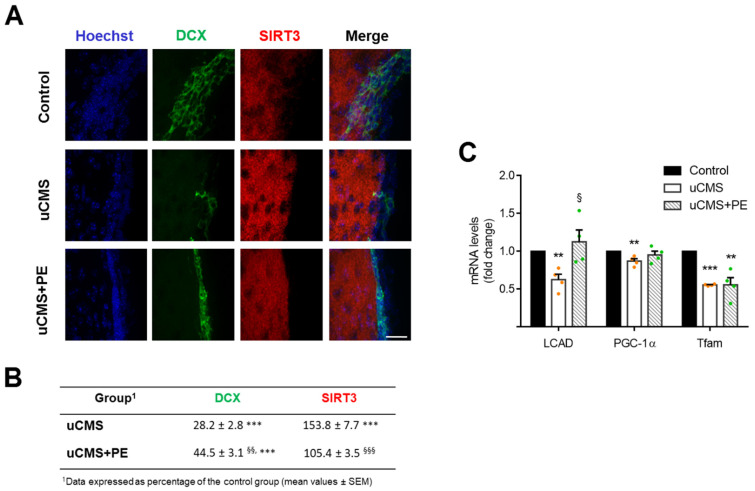
Reduced neurogenesis in depressive mice is associated with LCAD downregulation. Total RNA from SVZ-enriched extracts was processed for qRT-PCR and brain slices were processed for immunofluorescence analysis, as described in Section 2. (**A**) Representative confocal images of immunostaining for DCX (green, early neuronal differentiation marker) and SIRT3 (red) from SVZ. Nuclei were counterstained with Hoechst 33258 (blue). Scale bar, 20 µm. (**B**) Quantitative results of DCX and SIRT3 fluorescence signals from SVZ. Data expressed as percentage of the control group (non-depressed/unstressed animals). Data represent mean values ± SEM for at least ten data points per group. (**C**) Effect of physical exercise (PE) on LCAD, PGC-1α, and Tfam mRNA levels of uCMS mice. *Hprt* was used as loading control. Data are expressed as fold change over control group. Data represent mean values ± SEM for four individual mice per group (details in Section 2). ** *p* < 0.01 and *** *p* < 0.001 compared to control, ^§^
*p* < 0.05, ^§§^
*p* < 0.01 and ^§§§^
*p* < 0.001 compared to depressive animals (uCMS). Abbreviations: DCX, doublecortin; LCAD, long chain acyl-CoA dehydrogenase; PGC-1α, peroxisome proliferator-activated receptor γ coactivator α; tBHP, tert-butyl hydroperoxide; Tfam, mitochondrial transcription factor A; uCMS+PE, depressive mice undergoing physical exercise (PE).

**Table 1 cells-11-00090-t001:** List of primers used for qRT-PCR.

Gene	Sequence (5′-3′)
*Hprt*	5′ -GGTGAAAAGGACCTCTCGAAGTG- 3′ (fwd)
	5′ -ATAGTCAAGGGCATATCCAACAACA- 3′ (rev)
*Mki67 (for ki67)*	5′ -CCTTTGCTGTCCCCGAAGA- 3′ (fwd)
	5′ -GGCTTCTCATCTGTTGCTTCCT- 3′ (rev)
*Ppargc1a (for PGC-1α)*	5′ -GGACATGTGCAGCCAAGACTCT- 3′ (fwd)
	5′ -CACTTCAATCCACCCAGAAAGCT- 3′ (rev)
*Map2*	5′ -GTTCAGGCCCACTCTCCTTC- 3′ (fwd)
	5′ -CTTGCTGCTGTGGTTTTCCG- 3′ (rev)
*Tfam*	5′ -CACCCAGATGCAAAACTTTCAG- 3′ (fwd)
	5′ -CTGCTCTTTATACTTGCTCACAG- 3′ (rev)
*PLIN2*	5′ -TGCTGTGTGGTGATCTGGAC- 3′ (fwd)
	5′ -CGGAGGACACAAGGTCGTAG- 3′ (rev)
*VLCAD*	5′ -CAGCGACTTTATGCCAGGGA- 3′ (fwd)
	5′ -TGGCAGGGTCATTCACTTCC- 3′ (rev)

## Data Availability

The data supporting the findings in this study are available from the corresponding author upon reasonable request.

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
