# Peer review of "The Mitochondrial Antioxidant Sirtuin3 Cooperates with Lipid Metabolism to Safeguard Neurogenesis in Aging and Depression"

_cells, 2021, doi:10.3390/cells11010090_

Round 1

Reviewer 1 Report

In this manuscript, Só nia Sá Santos et al report the molecular mechanisms underlining SIRT3 cooperates with lipid metabolism to regulation of NSC fate in ageing and depression. The data are generally interesting and compelling. However, there are still a lot of concerns as noted below.

1. The manuscript repeatedly mentioned that the importance of lipid metabolism activated by SIRT3 for the prevention of NSC aging and proper differentiation of NSC, but only proved the interaction and expression of SIRT3 and LCAD, did not perform the detection of lipid metabolism.

2. Before exploring the tert-butyl hydroperoxide induces ageing features in NSCs and reduces mitochondrial antioxidant defence system, it is suggested to test stem cell markers and the capacity of tripotential differentiation mentioned in the manuscript.

3. In figure 1A, you only provide the photos of β-galactosidase droplets, while no statistical analysis. In addition, the protein expressions of P16, P21… in figure 1A, C, D lack immunoblot bands, only with the results of statistical analysis.

4. After overexpression of SIRT3 in Figure 2, there was also no statistical analysis in protein expression. In the final conclusion of result 2, it was concluded that SIRT3 directs towards NSCs to differentiation, but the only marker of differentiation in the manuscript was the detection of MAP2 (neuronal marker). It is suggested to add the other markers or experiments, such as immunofluorescence. Additionally, you chose NeuN as neuronal marker in Figure 1. What is the purpose of different markers used in your experiments? What’s more, the legend in Figure 2 is not very clear.

5. In Figure 4, after co-transfection of SIRT3 overexpression and siRNA LCAD, besides neuronal marker-MAP2 and aging related marker-SA-β-gal, there were no relative detection about mitochondrial energy and lipid metabolism. Moreover, in Figure 4B and 4C, the images (4 groups) and statistical analysis (6 groups) are inconsistent.

6. In Figure 5A and 5B, the images (3 groups) and statistical analysis (2 groups) are not consistent. 

Author Response

Response to Reviewer: 1

(Sá Santos et al.; manuscript 1471567)

We are thankful for the thorough review of the manuscript and are delighted that the Reviewer considered our data “interesting and compelling”, giving us the opportunity to improve certain aspects of the manuscript. However, he/she raised a number of questions and several concerns that we have minutely addressed in a revised version of the manuscript. By doing so, we feel that we are submitting an improved version of our studies that we hope is deemed acceptable for publication in Cells - Special Issue: Mitochondrial Functions in Stem Cells.

  1. The Reviewer is absolutely correct in this criticism. Following the Reviewer’s suggestion, we have re-written certain sentences of the manuscript to avoid overinterpretations on the role of lipid metabolism in SIRT3-mediated effects. Nevertheless, the importance of lipid metabolism on SIRT3 effects was now further validated by new data present in Figure 4A. This new set of experiments showed that Perilipin 2 (PLIN2), a cytosolic lipid droplet binding protein, is significantly decreased in aged NSCs, being only prevented by SIRT3 when LCAD is not silenced. These results reinforce the role of lipid metabolism in SIRT3-mediated effects on NSCs and were now discussed in the revised version of the manuscript.

  1. The Reviewer raised an important issue. However, since the capacity of tripotential differentiation of CGR8-NS cells was already tested in included in the previous article Glaser et al 2007 (new Reference 41 of the revised manuscript), we decided to not include these data in the present manuscript.

  1. Following the Reviewer’s recommendation, we have now included the statistical analysis of β-galactosidase droplets, as well as the representative immunoblot bands of p16, p21 and p53 in Figure 1A.

  1. The Reviewer is absolutely correct in this criticism. The statistical significance of SIRT3 overexpression was now included in the revised version of the manuscript. The reason why we selected different neural markers in Figures 1 and 2, relied on the laboratory logistics. In fact, we had available the primers for MAP2 and a specific antibody to NeuN. Since both markers are present in the same maturation stage of neurons, we through that the evaluation of other neuronal marker after co-modulation assays would further strength the emerging role of SIRT3 in regulating NSC differentiation potential. Nevertheless, we would like very much to improve this point in our manuscript by performing an immunocytochemistry against NeuN in co-modulation experiments. Unfortunately, as we previously explained to the Editor, we have been facing a yet unsolved technical problem on the cell culture room that has made impossible to perform this new experiment. In case we have the chance to conclude the experiments before proof article corrections, we would appreciate very much to have the opportunity to include the new images in the final manuscript.

  1. To address the Reviewer’s concern regarding the involvement of other lipid metabolism marker during SIRT3-rescued NSC differentiation, new RT-PCR experiments were performed and the mRNA levels of Perilipin 2 (PLIN2), a cytosolic lipid droplet binding protein, was assessed under the same cell context. As stated above, our results revealed that expression of PLIN2 was significantly decreased in aged NSCs. SIRT3, in turn, was capable of preventing PLIN2 decline, but only in the presence of LCAD. These results reinforce the role of lipid metabolism in SIRT3-mediated effects on NSCs. New data have been included in Figure 4A and discussed in the revised version of the manuscript. At last, in Figure 4, the representative images for the two missed groups in panel B were now added to be consistent with the statistical analysis in panel C.

  1. Although it is mentioned in the legend of Figure 5, panel B, the authors agree that some doubts may arise about the data values in the table. Therefore, a sentence similar to the one in the legend was added as a footer for clarification.

Reviewer 2 Report

In the manuscript by Santos et. al., the authors investigate the role of Sirtuin (Sirt)3 in neural stem cell ageing and differentiation using tert-butyl hydroperoxide. They have further extended their findings in the depressive-like behavior mice model using uCMS to investigate the functional significance of Sirt3. The authors have identified that tBHP regulates NSC ageing by altering mitochondrial oxidative damage and differentiation potential which are rescued by SIRT3. They have further demonstrated an interaction between SIRT3 and LCAD which is required for the rescue process. The study is interesting as the process of NSC differentiation has been studied in the context of aging. The role of LCAD in NSC proliferation has been demonstrated earlier by the same group and the regulation of LCAD by SIRT3 has also been reported independently (20203611). Additionally, the importance of SIRT3 in improving microglia activation-induced oxidative stress injury through mitochondrial apoptosis pathway in NSCs has also been studied earlier. Although the basic findings of the study have been identified and demonstrated by various research groups earlier, the present study may still provide a novel intervention target for NSC survival. However, there are a few concerns that must be addressed before consideration for publication.

1.The rationale for choosing SIRT3 among all SIRTs for this study is not clear. Also, are other SIRTs regulated in neurogenesis during ageing and depression?

2.In Fig1A, the quantitation of p16, p21, and p53 is shown. The authors should include the representative immunoblots.

3.It is not clear if Fig 1D represents protein or mRNA expression of SIRT3. The authors must mention it clearly in the graph/legend.

4.The authors should also check the activity of SIRT3 protein in tBHP-treated NSCs.

5.In Fig 2C, is the comparison of tBHP-treated cells significant with SIRT3+tBHP-treated cells?

6.In Fig3A, the inputs for the immunoprecipitaion experiment are missing and must be included to confirm equal initial input.

7.In fig 3, the authors must mark the band of interest as there are multiple bands detected on the blot.

8.The authors can include the lipidomics data to show the effect of SIRT3-mediated LCAD control of ageing in NSCs to strengthen the data.

9.Can the authors explain the discrepancy in the levels of MAP2 mRNA levels in the Fig 2B and 4A for tBHP+SIRT3-treated cells? The trend is inverse in both the figures.

Author Response

Response to Reviewer: 2

(Sá Santos et al.; manuscript 1471567)

We are pleased that the Reviewer considered our manuscript interesting and novel.  However, the Reviewer considered that few concerns that must be addressed before consideration for publication.  Therefore, we have performed additional statements and made suggested modifications that we included in the revised version of the manuscript.  By doing so, we feel that we are submitting an improved version that we hope fulfills the standard requirements for publication in Cells - Special Issue: Mitochondrial Functions in Stem Cells.

  1. The Reviewer’s remark is extremely relevant. In fact, it is possible that other SIRTs, such as SIRT1 and SIRT2, might also regulate neurogenesis, aging (Ng et al. Cell. Neurosci. 2015) and even depression (Liu et al., Scientific Reports 2015). However, since the scope of our study was to further understand the impact of mitochondrial metabolism on NSC plasticity under several scenarios of NSC dysfunction, we decided to only focus on SIRT3, a central player in mitochondrial metabolism and oxidative protection. This relevant information was now further explained in the Discussion section of the revised manuscript.

  1. Following the Reviewer’s suggestion, we have now included the representative immunoblots for p16, p21 and p53 in Figure 1A.

  1. We apologize for the inconvenient of not clarifying the graph/legend in Figure 1D. It is now clarified on the graph axis “SIRT3 protein levels”, in accordance what was originally stated in the legend.

  1. and 5. Regarding the Reviewer concern on the activity of SIRT3 protein in tBHP-treated NSCs, we showed in Figure 2D that mtROS levels were significantly reduced in aged NSCs overexpressing SIRT3, when compared with aged NSC alone. In fact, although the acetylation levels of SOD2 appeared to not change between these two conditions, when normalized to total SOD2 (Figure 2C), it is possible that either SOD2 turnover or its fluctuation levels do not allow the detection of SOD2 deacetylation by SIRT3. Nevertheless, we believe that the conclusive read-out for SIRT3 activation in aged NSCs is expressed in Figure 2D, in which the downstream effect of SOD2 activation were measured.

  1. The Reviewer is absolutely correct in this criticism. We have now included a supplementary Figure to show the initial input for the immunoprecipitation experiments.

  1. As suggested by the Reviewer, a red arrow was now added to mark the band of interest in Figure 3A and B.

  1. Although exploring possible changes in lipidomic upon SIRT3-mediated aged NSC is of upmost importance, we feel that it falls outside the scope of this manuscript and would be best suited for future studies. Nevertheless, this possibility was addressed and discussed in the Discussion section of the revised manuscript.

  1. We understand the Reviewer’s concern regarding the discrepancy in the mRNA levels of MAP2 in Figure 2B and 4A for tBHP+SIRT3-treated cells. We attribute this difference to the fact that in Figure 4, all cells, even those not downregulated for LCAD, were subjected to a higher dose of lipofectamine (co-modulation of SIRT3 and LCAD), when compared to cells in Figure 2 (overexpression of SIRT3 only). Indeed, although the data between both Figures are not comparable, we could appreciate that under low levels of LCAD (assessed by siRNA LCAD), the SIRT3 is no longer capable of abrogating aged-induced MAP2 decline.

Round 2

Reviewer 1 Report

The authors have addressed most of my concerns. However, two points remain.

  1. The detection of MAP2 mRNA expression alone is not sufficient to explain the differentiation potential of NSCs.  
  2. In Figure 4, after co-transfection of SIRT3 overexpression and siRNA LCAD, the authors added a new RT-PCR assay to assess the mRNA levels of cytoplasmic lipid-droplet binding protein Perilipin 2 (PLIN2) in the same cellular environment. However, mitochondrial oxidative stress should also be detected, such as mitochondrial ROS and acetyl-SOD2/SOD2 levels.

Author Response

Response to Reviewer: 1

(Sá Santos et al.; manuscript 1471567)

We thank the Reviewer for his comments on the revised manuscript. Regarding the concerns raised by the reviewer, we would like to highlight the following:

  1. and 2. Although we are confident that our results are robust enough to support the conclusions made, we understand that evaluation of other neuronal and glial protein levels would certainly strengthen the conclusion that SIRT3 rescues the differentiation potential of aged NSCs. In addition, detecting mitochondrial ROS in cells undergoing co-modulation experiments would be interesting to corroborate the effect of SIRT3-mediated SOD2 activation in abrogating NSC dysfunction. As we previously mentioned, we are facing a yet unsolved technical problem on the cell culture room that enables to perform those experiments at this moment. In addition, due to the COVID-19 Contingency plan, our Institution will be forced to close until January 10, 2022 (https://www.bloomberg.com/news/articles/2021-11-25/portugal-says-remote-working-to-be-mandatory-in-week-of-jan-2-9), turning impossible to solve the technical issue and pursue the required experiments in a reasonable time frame. Therefore, we have attempted to discuss the drawbacks of our study and highlight future directions. By doing so, we are confident that we are submitting the most feasible upgraded version of the article at this moment.

Reviewer 2 Report

The authors have addressed all the comments by either providing the data or justifications. I believe providing the lipidomics data would have strengthened the manuscript, however the authors have a plan of persuing this data for future work and it has no impact on findings of the manuscript. The justifications given for the raised concerns are convincing and revised manuscript should be accepted for the publication.   

Author Response

Response to Reviewer: 2

(Sá Santos et al.; manuscript 1471567)

We are pleased that he/she considered the manuscript surely improved from the previous version and with the standard requirements for publication in Cells, Special Issue: Mitochondrial Functions in Stem Cells.